# Technical Note: Pyrolysis principles explain time-resolved organic aerosol release from biomass burning

Mariam Fawaz<sup>1</sup>, Anita Avery<sup>2</sup>, Timothy B. Onasch<sup>2</sup>, Leah R. Williams<sup>2</sup>, and Tami C. Bond<sup>1,3</sup>

<sup>1</sup>Department of Civil and Environmental Engineering, University of Illinois Urbana-Champaign, 205 N Mathews Ave, Urbana, Il 61801

<sup>2</sup>Aerodyne Research Inc., Billerica, Massachusetts 01821, United States

<sup>3</sup>Department of Mechanical Engineering, Colorado State University, 400 Isotope Dr, Fort Collins, CO 80521

Correspondence: Tami C. Bond (tami.bond@colostate.edu)

Abstract. Emission of organic aerosol (OA) from wood combustion is not well constrained; understanding the governing factors of OA emissions would aid in explaining the reported variability. Pyrolysis of the wood during combustion is the process that produces and releases OA precursors. We performed controlled pyrolysis experiments at representative combustion conditions. The conditions changed were the temperature, wood length, wood moisture content, and wood type. The mass loss

- 5 of the wood, the particle concentrations, and light gas concentrations were measured continuously. The experiments were repeatable as shown by a single experiment, performed nine times, in which the real-time particle concentration varied by a maximum of 20%. Higher temperatures increased the mass loss rate and the released concentration of gases and particles. Large wood size had a lower yield of particles than the small size because of higher mass transfer resistance. Reactions outside the wood became important between 500 and 600°C. Elevated moisture content reduced product formation because
- 10 heat received was shared between pyrolysis reactions and moisture evaporation. The thermophysical properties, especially the thermal diffusivity, of wood controlled the difference in the mass loss rate and emission among seven wood types. This work demonstrates that OA emission from wood pyrolysis is a deterministic process that depends on transport phenomena.

#### 1 Introduction

The emission of organic aerosol (OA) particles from open biomass and biofuel burning contributes 93% of the global atmo-15 spheric primary organic aerosol emissions (POA), with 19% from biofuel burning for residential energy consumption (Bond et al., 2004). OA emission factors from biofuel burning are used in emission inventories (Denier van der Gon et al., 2015; Wu et al., 2019; Morino et al., 2018; Cao et al., 2006), regional and global air quality models (Lane et al., 2007; Yu et al., 2004; Wang et al., 2018; Tsimpidi et al., 2017; Theodoritsi and Pandis, 2019), and exposure studies for health impacts (Tuet et al., 2019; Hystad et al., 2019; Carter et al., 2017; Okello et al., 2018).

Emission of OA from biomass is observed to be variable and chaotic, making the determination of definite emission factors difficult (Nielsen et al., 2017; Shrivastava et al., 2006; Jolleys et al., 2014). To explain measurement variability, the emission behavior has been related to combustion parameters. The most used parameters are heat flux (Haslett et al., 2018; Eriksson et al., 2014), wood type (Weimer et al., 2008; McDonald et al., 2000) whether softwood or hardwood (Schmidl et al., 2008;

Inuma et al., 2007), moisture content (van Zyl et al., 2019), and combustion phase whether flaming or smoldering (Yokelson 25 et al., 1997; McKenzie et al., 1995; Amaral et al., 2014). The most prevalent characterization factor in emission studies is the modified combustion efficiency (MCE) (Ward and Hao, 1991); others, such as the fire radiative energy (FRE) (Freeborn et al., 2008; Ichoku and Kaufman, 2005) and inferred pyrolysis temperature (Sekimoto et al., 2018) have also been used. Previous emission studies have alluded to the role of pyrolysis in the emission of organic aerosols (Yokelson et al., 1996; Sekimoto et al., 2018); however, there has not yet been a comprehensive attempt to understand how emission occurs and how governing factors at the point of origin affect the types and amounts of particles and gases. We suggest that the study should start at the 30 source of emission.

When wood is heated, the first response is thermal degradation, known as pyrolysis. Pyrolysis reactions form products in the solid phase (char) (Shafizadeh, 1982), in the gas phase as permanent gases (e.g, CO, CO<sub>2</sub>, and hydrocarbons) and as condensable products (CP) that have high molecular weight (Evans and Milne, 1987). The latter are known as tar in pyrolysis

- literature, and these products overlap with condensed organic aerosol. Owing to the pressure gradient between the formation 35 site and the surface, both gas and CP (Diebold, 1994; Suuberg et al., 1996) can be ejected from the wood (Staggs, 2003). Before release, CP can be broken down to form gaseous material (Pattanotai et al., 2013) due to secondary pyrolysis reactions within the wood. After release and in the absence of oxidation reactions associated with combustion, the CP may condense to become particles (Zhang et al., 2013), or they may crack at high temperature to form permanent gases (Boroson et al., 1989b;
- Morf et al., 2002) due to secondary reactions outside the wood. After ignition, gas phase oxidation reactions can consume CP to form permanent gases that will not condense at any relevant temperature, such as CO and CO<sub>2</sub>. CP may also polymerize or undergo at high temperatures to form soot (Atiku et al., 2017; Fitzpatrick et al., 2009). All of these processes reduce the yield of OA. The exothermic reactions that occur outside or at the surface of the wood, often called flaming and smoldering, provide energy for the pyrolysis of unreacted wood and production of OA precursors.
- Considering the processes that control emission can lead to a greater understanding of the connection between OA emission and the nature of the fuel and the combustion process. A systematic sequence of investigation would evaluate several processes: (a) The control of solid phase processes, or pyrolysis, on the emission of CP and gases and the potential of CP to become OA; (b) The alteration of OA emission caused by the gas phase processes oxidation (including flaming), polymerization, and secondary pyrolysis; (c) The feedback energy the solid phase processes receive from the heat release from gas phase oxidation reactions.

Much of the work in pyrolysis literature aims to describe industrial applications such as bio-oil production and energy generation from biomass. "thermally-thin" applications employ pulverized wood for which heat transfer to the center occurs nearly instantly (Wagenaar et al., 1993; Di Blasi and Branca, 2001; Janse et al., 2000). Product yield from thermally-thin wood is measured either as lumped groups of tar, char and gases (Di Blasi et al., 2001b; Grønli and Melaaen, 2000) or speciated

measurements of gaseous molecules and tar molecules using FTIR or GC-MS (Anca-Couce et al., 2017; Corbetta et al., 2014; Dufour et al., 2007). These studies can provide final yields for industrial applications, product types, and kinetic parameters for modeling thermally-thick pyrolysis.

In thermally thick pyrolysis, heat transfer in and mass transfer out of the wood matrix are important limiting factors. This type of pyrolysis is less often explored because of its limited relevance to industrial processes, although decomposition rates of thermally thick wood have been examined because they affect fire safety (Tran and White, 1992; Lee et al., 1977). Samples investigated are commonly 1-3 cm, with cylindrical and spherical shapes (Remacha et al., 2018; Bennadji et al., 2013; Di Blasi et al., 2001b; Ding et al., 2018b; Gauthier et al., 2013). Differences in time-dependent mass loss and in overall product yield occur over a range of wood sizes, shapes, and temperature or heat flux to which they are exposed. Throughout these works, larger sizes and non-rounded shapes have not been investigated to ascertain whether known pyrolysis principles are sufficient to explain behavior in real-world applications. Furthermore, time-dependent release occurs because of heat and mass transfer

- to explain behavior in real-world applications. Furthermore, time-dependent release occurs because of heat and mass transfer rates within the wood matrix. A few of the studies referenced above reported time-resolved escape of light gases, but none have measured time-resolved emission of condensable products, so there is little evidence to indicate whether the product division changes as the pyrolysis front penetrates into the wood. Any such shifts could affect both ignition timing and emission characteristics.
- The name pyrolytic organic matter (PyOM) is introduced here as the name of particles that are directly emitted from pyrolysis in the absence of oxidation reactions inside and outside the wood. In literature, common names of organic aerosol particles from biomass burning include biomass burning organic aerosol (BBOA), brown carbon (BrC), primary organic aerosols (POA), and secondary organic aerosols (SOA), but these products are not exactly like pyrolysis particles in their properties or emission behavior. BBOA contains oxidation products from combustion and atmospheric oxidation reactions (Huffman et al., 2009).
- BrC is defined based on the optical properties of OA (Andreae and Gelencsér, 2006). POA and SOA from biomass burning come from all combustion processes, fresh (May et al., 2013), or aged (Capes et al., 2008), and sampled at the source or in the atmosphere (Cubison et al., 2011).

In this work we developed an approach that relates the emission measurements from wood pyrolysis with the processes responsible for the emissions in the wood. This approach was accomplished by coupling our previously developed pyrolysis

model (Fawaz et al., 2020) with emission measurements. The objective of the work presented here is to determine the extent to which released PyOM is governed by transport phenomena within the wood during pyrolysis. This is achieved by selecting and varying the important factors that influence heat, mass, and momentum transfer in pyrolysis: thermal boundary conditions, wood size, moisture content, and wood type. Chemical characterization of emitted PyOM was also made and will be discussed in future work.

# 85 2 Methods

This section describes the experiments performed to measure the change in the mass of the wood and the release of gas and particles during controlled pyrolysis. The details of the pyrolysis model used in the analysis of the experimental results was published in Fawaz et al. (2020).

# 2.1 Sampling Setup

- Figure 1 is a schematic of the experiment. The pyrolysis reactor was described in detail in previous work (Fawaz et al., 2020) and is briefly reviewed here. The reactor was made of a 2.6 kW cylindrical heater controlled by a PID temperature controller. Nitrogen gas flowed through the reactor to maintain oxygen deficient conditions and a precision scale measured the wood mass change.
- Smoke emitted from the reactor cooled during the primary dilution stage in the fume hood and was pulled into a duct at
  a speed of 7 m/s. Gases were sampled directly from the duct and the particles were measured after undergoing a secondary dilution stage in a probe. The secondary dilution probe was 1 m in length and had two concentric tubes (OD = 0.75 in, ID = 0.5 in); the internal tube was made of a porous metal with a 20µm pore size. The sample flow entered the inner tube, and the dilution flow entered the outer tube. The flow of air from the outer tube to the inner tube formed a barrier to reduce the loss of particles to the walls of the probe. Compressed HEPA and activated carbon filtered air was provided to the dilution probe.
  The secondary dilution ratio was the ratio of the dilution flow of air to the sample flow from the duct. Air and vacuum flows
- were monitored by two low pressure drop mass flow meters (Alicat, MW-100SPLM/175SPLM), and the sample flow was measured directly using a bubble flow meter before and after each experiment. The secondary dilution ratio across experiments ranged between 150 and 200 (Table S1). A dark, heavy, sticky, material was released in some experiments, especially at lower temperatures. This material deposited on the cold surfaces of the reactor before reaching the sampling system and could not be
   collected.

The number size distribution of particles was measured by an Engine Exhaust Particle sizer (TSI, EEPS 3090), the mass was measured by a DustTrak Aerosol Monitor (TSI, DustTrak 8530), the concentrations of CO and  $CO_2$  gas were measured by a CO/CO<sub>2</sub> analyzer (Horiba, AIA-220), and the total concentration of carbon content in gas phase hydrocarbons (HC) was measured by a flame ionization analyzer (Horiba, FIA-236). The gas instruments were calibrated every day with zero and span

gas. The electrometers of the EEPS were zeroed every day to check the stability of the instrument.

# 2.2 Wood Sample Description

Wood samples of varying type, size, and moisture content were used. Seven types of wood were used: Douglas fir (Pseudotsuga Menziesii,  $\rho = 587 \pm 10 \text{ kg.m}^{-3}$ ), two types of pine (Pinus Radiata, termed pineR,  $\rho = 507 \pm 4 \text{ kg.m}^{-3}$  and Pinus Echinata, termed pineE,  $\rho = 538 \pm 11 \text{ kg.m}^{-3}$ ), ipe (Handroanthus,  $\rho = 937 \pm 13 \text{ kg.m}^{-3}$ ), maple (Acer Nigrum,  $\rho = 747 \pm 17 \text{ kg.m}^{-3}$ ), birch (Betula Papyrifera,  $\rho = 686 \pm 11 \text{ kg.m}^{-3}$ ), and poplar (Liriodendron Tulipifera,  $\rho = 554 \pm 20 \text{ kg.m}^{-3}$ ). The end grain and

birch (Betula Papyrifera,  $\rho = 686 \pm 11 \text{ kg.m}^{-3}$ ), and poplar (Liriodendron Tulipifera,  $\rho = 554 \pm 20 \text{ kg.m}^{-3}$ ). The end grain and flat grain of each wood type are shown in Fig S1. The apparent density of each wood sample was calculated as the ratio of the

mass to the volume of the wood sample. PineR, pineE, and Douglas fir are softwoods and ipe, maple, birch, and poplar are hardwoods. In lumber, the division of softwood and hardwood can be based on botanical or anatomical definitions. Botanically, hardwoods belong to the angiosperm

family and softwoods to the gymnosperm family, and anatomically, hardwoods are more porous than softwoods (Bergman et al., 2010). Chemically, the difference between softwood and hardwood is less clear (Di Blasi et al., 2001a). The main

structural groups of wood are cell wall material, extractives, and ash. Hemicellulose, cellulose and lignin are the components of cell wall material, volatile sugar and acids are the major components of extractives, and inorganic oxides make up ash (Reed, 2002).

- Two size dimensions of wood were used; 14 cm x 3.8 cm x 2.9 cm are called in this work "large", and 2.9 cm x 2.9 cm x 2.9 cm are called "small". The wood was cut longitudinally with the the grain as shown in Fig S2. The large wood size was the same order of magnitude as those used in wood cookstoves, and the small size was used to evaluate the effect of the length of the wood on emission. Both wood sizes are considered thermally thick, a condition in which a thermal gradient forms between the surface and the center when the wood is exposed to heat from its surroundings (Pyle and Zaror, 1984).
- Two wood moisture contents (MC) were used for pineR: dry samples at  $8\pm1\%$  and wet wood at  $25\pm2\%$ . Dry wood samples used were received as kiln dried wood and the MC of the samples was between 7-9%. The wood MC was increased by soaking pineR wood samples for 24 hours based on the method of Lee and Diehl (1981). The MC of the soaked wood was measured in four internal positions to confirm the permeation of water and a homogeneous distribution of water throughout the wood.

## 2.3 Experimental Design

Two groups of experiments were performed. The first group varied pyrolysis conditions of reactor temperature, wood moisture content and wood size for two wood types: birch and pineR. PineR is a low-density softwood and birch is a high-density hardwood. These two woods were chosen to evaluate whether the response to changing input conditions depends on wood type. The second group of experiments explored how emissions from pyrolysis change based on the type of the wood. Seven wood types, including Birch and PineR, were used in constant temperature experiments at 400, 500, and 600°C. The conditions of all the experiments used in the analysis are shown in Table S1.

# 2.4 Reactor Operation

The experiments were performed under isothermal conditions; the wall temperature of the reactor was maintained at a set point temperature of either 400, 500, or 600°C. To begin an experiment, the reactor was raised to the set point temperature, after which data collection was initiated and nitrogen gas purged the reactor for 2 minutes at 20 LPM. The wood sample was inserted after the purge period, and this event marked the experiment start time.

After the wood sample was inserted, the reactor was kept partially covered with firebricks to prevent heat loss to the ambient atmosphere. The experiment ended when the residual mass of the solid stopped changing within the uncertainty limit of the mass balance and when the hydrocarbon gas concentration returned to background values measured before the experiment. Each experiment was replicated to determine the uncertainty.

## 150 2.5 Analysis

A LabVIEW program collected the thermocouple, mass balance, gas analyzer, and flowmeter data. TSI softwares were used to collect data from the EEPS and DustTrak, and the data streams were synchronized with the signals collected in LabVIEW.

The background concentration of the gases in ambient air was measured before and after the experiment and subtracted from the gas signal during the experiment. We assumed a linear drift in the signal between initial and final measurements. The real-time concentrations were smoothed using a MATLAB moving mean function with an averaging period of 3 secs, unless otherwise stated. Product collection was calculated as the total mass of all measured products divided by the initial mass of the wood. The products measured include the final char mass and the total mass of PyOM, CO, CO<sub>2</sub>, and HC. The yield or mass fraction of each product is the sum of its real-time measured mass divided by the initial mass of the wood. In the elevated MC experiments, calculated mass fractions are provided on an as-received basis and wet basis, where the as-received mass of the wood was the mass before water absorption and the wet mass was after absorption, respectively.

The EEPS measured the number concentration of the particles from 6 - 560 nm. A density of  $1.3*10^3$  kg.m<sup>-3</sup> was used (Hennigan et al., 2011) to convert from number to mass distribution. EEPS data were corrected for counting errors; details of the correction are in section S2 and summarized here. The EEPS underestimates the size of large particles (D<sub>p</sub>>100 nm) (Zimmerman et al., 2014), so the method of (Lee et al., 2013) was used to correct size distributions by comparing measured concentrations for size-selected ammonium nitrate particles between the EEPS and a long time-of-flight aerosol mass spectrometer

```
```

(LToF-AMS, Aerodyne Research, Inc.).

Pyrolysis experiments at 400°C released particles with diameters larger than the upper limit of the EEPS measurement range, causing an underestimation in the mass concentration. When the particle sizes exceeded the size range measured by the EEPS, the mass concentration measured by the DustTrak was used for the total mass of the particles. In section S3, we compare the

170 PyOM concentrations from the EEPS and DustTrak measurements at 400 and 500°C to evaluate the differences between the two instrument outputs and the adequacy of using the DustTrak data at 400°C.

The measured particle concentration was multiplied by the dilution ratio of the secondary dilution stage during the experiment. Thus, the PyOM mass concentrations reported here are those that would be measured in the primary dilution stage, so they are directly comparable to the gas concentrations.

# 175 3 Results and Discussion

Table 1 shows the yields of all measured products for the two groups of the pyrolysis experiments. The yields of PyOM show that 10-30% of the initial wood mass was emitted as particles, depending on the conditions of the experiment. One test (large birch at 500°C) was repeated nine times, and the average coefficient of variation of the real-time mass concentration of PyOM was  $15\pm3\%$  (Fig S21), indicating that the tests were repeatable. Collected product percentages, excluding water, of all

180 experiments were between 77% and 90% of the wood mass. The lower end of the mass closure was observed at 400°C because of the loss of dark heavy sticky material onto the reactor. The assumption of the PyOM density did not have a quantified effect on the PyOM yield and mass closure.

#### 3.1 Effect of Temperature

- Figure 2 shows the real-time mass loss rate and mass concentration of emitted gases and PyOM during pyrolysis of large pineR
  and birch wood at 400, 500, and 600°C. The time series all have two peaks, although some peaks are more pronounced than others. At each fixed temperature, the mass loss rate, PyOM, and gas concentrations exhibit similar behavior with time but the relative heights of the two peaks differ among temperatures. Figure S23-S22 plot the mean and standard deviations of the experiments discussed in this section; they show that the observed trends are caused by the wood response to the boundary conditions and are not experimental artifacts.
- Using a pyrolysis model verified experimentally, Fawaz et al. (2020) showed that the real-time mass loss rate from thermally thick wood could be predicted, and that internal and external heat transfer govern the thermochemical degradation that produces the gases and PyOM precursors. That work, and other pyrolysis literature of thermally thick wood (Di Blasi et al., 2000; Ding et al., 2018a), show that the two distinct peaks of mass loss rate that appear in Fig 2a-c can be explained by the reactor temperature and thermal diffusivity of a given wood type at a fixed size. Modeling results of the mass loss rate of birch and pineR at 400, 500, and 600°C are shown in section S4.2.
  - The heat the wood receives at the surface initiates reactions and is conducted towards the center of the wood. The surface layers of the wood gain temperature rapidly, reacting and losing mass to form the first peak in emissions. When these layers are depleted the declining portion of the first peak forms. Each location inside the wood receives heat conducted from the surface, and transfers heat toward the wood center. As pyrolysis progresses, the outer layers of the wood become char, which
- has a lower thermal conductivity than wood. This thermal resistance barrier slows the travel of the elevated temperature zone, creating the portion of the mass-loss curve that appears as a saddle at 500 and 600°C. The rate of mass loss declines when most of the material has reacted in the heated section, and when unreacted wood is not heated rapidly enough to maintain the same level of product formation. This principle explains the sharp second peak, which occurs when the thermal wave reaches the center. The earlier peak times and shorter overall pyrolysis duration of pineR compared to birch at all temperatures is due
- to the lower density and greater thermal diffusivity of pineR compared to birch. The higher the thermal diffusivity the faster the heat transfer rate from the surface to the center, making pyrolysis reactions in pineR faster than birch. Birch has a larger overall mass loss; the higher density of birch compared to pineR means there is more mass to react and form products. Mass loss as a function of time and temperature is repeatable (Fig S21) because it is governed by the relative rates of heat transfer, temperature increase, and char formation within the wood.
- At all temperatures, the real-time mass concentration of PyOM (Fig 2d-f) had similar real-time features to the mass loss rate. The concentration of CO,  $CO_2$ , and HC (Fig 2g-o) are also similar in behavior. For example, the slope of the first peak increases with temperature for all measured products. Relative peak heights and the nature of the saddle between peaks changed with wood size and temperature in nearly the same way for mass loss rate, PyOM, and gases. The relative magnitude of PyOM and gases is relatively constant, regardless of the wood type and temperature, suggesting that the product emission throughout the
- entire pyrolysis process is similar. Concentrations of CO and  $CO_2$  at 400°C are noisy because they are near the instrumental detection limit.

Emissions of gases and PyOM increase as a function of increasing temperature, except for PyOM at 600°C (Figure 2f) for which the PyOM was lower than at 400 and 500°C. At and above this temperature, secondary pyrolysis reactions (known as tar cracking reactions in pyrolysis literature) break down CP and transform them to CO and CO<sub>2</sub> (Morf et al., 2002), and increase CO yields compared to CO<sub>2</sub> and other gases (Nunn et al., 1985). These reactions are likely the cause of the reduced PyOM yield at 600°C and the more than doubling in CO concentration from 500 to 600°C.

# 3.2 Wood size

At smaller wood sizes, heat transfer into the center takes less time, and pyrolysis and product release are completed more rapidly (Fig S30 and S31). If no reactions occur after pyrolysis, equivalent product yields would be expected from small and large wood. Transport through the porous char matrix after pyrolysis depends on the permeability of the material, the travel distance between the gas production site and the surface, and the pressure difference between the production site and the surface. If the precursors are not able to exit the wood freely, large and small wood might have different yields.

At 400°C, transport resistance inside the wood appears to reduce PyOM emission from larger wood, as the yield of PyOM for small birch and small pineR was higher than that for large wood (40% difference for birch and 15% difference for pineR).

- Yields of PyOM are shown in Table 1 and summarized graphically in Fig S32 and S33. These differences are not attributable to reactions outside the wood, where the residence time at elevated temperature was kept constant. Neither are they likely to be caused by product degradation within the wood (Boroson et al., 1989a, b). The influence of wood size in birch, along with a higher density that is associated with lower total porosity (Plötze and Niemz, 2011), caused a greater inhibition of mass release compared to pineR.
- Increased temperature generates gases more quickly, creating a greater pressure gradient and reducing the likelihood of mass transfer resistance. At 500°C, there was no difference in PyOM yields between large and small wood for pineR. However, at 500°C, mass transfer resistance still reduced the PyOM yield between large and small wood for the higher-density birch. There was no difference in yield between wood sizes for either type at 600°C.

## 3.3 Moisture content

- When free water evaporates from the wood matrix, less energy is available for thermochemical degradation and product release (Fatehi and Bai, 2014), and the rates of these processes are reduced. Figure 3 shows that the mass loss rate and real-time mass concentration of PyOM for pineR at MC=8% (dry) and at MC=25% (wet) have some similar features, such as the dual peaks and change in mass loss rate magnitude with reactor temperature. Many of the differences, including the time required to complete the reaction, can be explained by the energy consumed by evaporation.
- Compared with dry wood, initial mass loss from wet wood was equal or faster due to rapid evaporation of water at the surface. Evaporation occurs at 100°C and pyrolysis reaction rates become significant at temperatures higher than 280°C (Broido, 1976). The first peak results from the balance between radiative heat flux at the surface and the formation of char and is most affected by the change in relative rates. At 400 and 500°C, heat transfer into the wood is slow enough that evaporation at 100°C occurs before pyrolysis reactions at the surface and the first peak of PyOM emission is reduced to a broad shoulder appeared instead.

At 600°C, heat transfer is rapid and both pyrolysis and evaporation occur simultaneously at the surface. After the initial peak, continuous internal heat transfer in the wood sustains pyrolysis and product formation and there is little difference between the second peak height in the dry and wet cases.

The sharing of energy between water evaporation and pyrolysis shifts product yields towards char (Beaumont and Schwob, 1984; Peters and Bruch, 2003; Di Blasi et al., 2000) and reduces the reaction temperature. The effect of reaction temperature on yield reduction is evident in the change in yield between 400°C and 500°C where more char, less gases, and less PyOM

were produced at the lower temperature. Further, the yield of PyOM from pyrolysis of wet pineR on an as received basis was lower than that of pineR at MC=8% at the same reactor temperature (Table 1).

Published findings show conflicting effects of moisture content on particle emission yields. Some observe that higher moisture content increases OA emission (May et al., 2014; van Zyl et al., 2019; Magnone et al., 2016), while others find a reduction in OA emission with moisture content (Huangfu et al., 2014). Wood moisture has competing effects on emission. Wet wood

in OA emission with moisture content (Huangfu et al., 2014). Wood moisture has competing effects on emission. Wet wood has a higher ignition delay and a higher critical heat flux for ignition (Simms and Law, 1967), prolonging pyrolysis and CP emission that occur in the absence of oxidation processes. As shown here, the rate of condensable product formation itself may be diminished, and that could reduce the rate of oxidation reactions after ignition (Price-Allison et al., 2019).

# 3.4 Wood type

- Seven types of wood were used in pyrolysis experiments at each reactor temperature. Thermal properties of dry wood depend on density (MacLean, 1941), so we present results grouped by density, although there may be other causes of differences. Figures 4a and 4c show mass loss rate and the real-time mass concentration of PyOM from woods with densities below 600 kg.m<sup>-3</sup>, and Fig 4b and 4d shows the same quantities for woods with densities above 600 kg.m<sup>-3</sup>. The reactor temperature 500°C is shown because it is least sensitive to mass transfer resistance (low temperature effect) and secondary reactions (high temperature effect), as discussed previously. For each wood, regardless of density, its real-time PyOM mass concentration and
- mass loss rate share the same features in terms of pyrolysis duration, peak shape and relative peak heights.

Low-density wood types include softwoods (pineE, pineR, and Douglas fir) and one hardwood (poplar). These wood types had similar behavior, except for Douglas fir with a 20% longer pyrolysis duration. The peak heights in the mass loss rate and real-time mass concentration are within 25% of each other, showing that for this group the pyrolysis behavior is similar.

- The high-density wood types include birch, maple, and ipe, all hardwoods. The mass loss rate of the wood types in this group had different pyrolysis durations and ratios between the first and second peak compared to the low-density wood, with a shorter first peak and a larger second peak at the end. The higher second peak, a repeatable feature for high-density wood, is caused by the availability of more unreacted wood mass internally than in low-density wood. Ipe emitted more particles than birch and maple and had a higher yield even though the magnitude of the mass loss rates were comparable among the three wood types.
- This work's thermochemical approach does not offer the reason behind the high particle emission for ipe compared to other woods and remains an open question.

In pyrolysis studies, wood type effect has been distinguished in thermally thin wood based on the fractions of cellulose, hemicellulose, and lignin (Grønli et al., 2002). Di Blasi et al. (2001a) found that the mass loss rate behavior for thermally

and heating rates in these studies were not reported.

thick wood can be related to the chemical makeup at low heat flux, and at higher heat fluxes the mass loss rate was explained 285 by transport phenomena. When heat transfer controls pyrolysis and burning, the difference in the mass loss rate of the wood can be explained using the thermal diffusivity of the wood (Spearpoint and Quintiere, 2001). As demonstrated in this work and modeled (Fawaz et al., 2020), the mass loss rate change as a function of wood type and heat flux can be explained by the heat transfer in the wood. Section S4.2 shows that the mass loss rate of birch and pineR can be predicted by the same global kinetics using Gpyro (Lautenberger and Fernandez-Pello, 2009).

Measurements of emission factors to represent open biomass burning have often been reported for individual wood types (Stock-290 well et al., 2014; Chen et al., 2007). Tabulations for use in atmospheric models have been grouped by ecosystem type (Akagi et al., 2011; Andreae, 2019), sometimes separating by flaming and smoldering (Koppmann et al., 2005). Measurements of wood burning in fireplaces and domestic stoves have provided in-depth chemical composition of gas and particulate emission for different wood species (McDonald et al., 2000; Schauer et al., 2001; Vicente and Alves, 2018). However, these studies could not attribute variations of measured emission yields to wood type (Goncalves et al., 2011; Ozgen et al., 2014). Temperatures 295

We have shown that differences in yields of particles and gases could be explained by reactor temperature, moisture content, and wood size, and that time-dependent emission of products can be explained by differing thermal diffusivities rather than wood species. Varying PyOM and gas emissions from different vegetation types, as widely reported in literature, are likely attributable to these factors rather than to differences among wood species.

# 300

#### Conclusions 4

The results of the developed approach reported here show that time-dependent emission of particles and gases from pyrolysis is a repeatable and deterministic process. Coupled with model findings, we found that the real-time mass loss rate can be explained by heat and mass transfer processes, and in turn, real-time emission of particles and gases follows the mass loss rate.

305

The conditions of pyrolysis influence the product yield and the real-time concentrations. We reported here the effect of the heating conditions, wood size, moisture content, and wood type on product emission. Comparative yields of particles and gases can be explained by thermochemical principles that are well known in pyrolysis literature. Increasing temperatures increase mass loss rates and increase the concentration of emitted particles between 400 and 500°C. At 600°C the particle concentration decreased, likely due to gas phase secondary reactions outside the wood. Increasing wood size decreased the

yield of particles at the lowest temperature due to mass transfer resistance. Increasing moisture content reduced the yield and 310 real-time concentration of particles and gases when drying reactions consumed some of the energy that the wood received for pyrolysis. The mass loss rate of different wood types showed differences between low-density ( $\rho < 600 \text{ kg.m}^{-3}$ ) and high-density wood ( $\rho > 600 \text{ kg.m}^{-3}$ ).

We have demonstrated that the first step in biomass emission—release of particles and gases from pyrolysis— is predictable

and repeatable. The principles demonstrated here have not previously been exploited to guide understanding of biomass-315 burning emissions to the atmosphere. Quantifying relevant physical factors, such as temperature, wood size, and thermal