# Peer review of "Technical Note: Pyrolysis principles explain time-resolved organic aerosol release from biomass burning"

_Atmospheric Chemistry and Physics, 2021_

## Referee Comment (RC1)

The paper addresses an important issue in understanding the reported variability of emission of organic aerosol (OA) from wood combustion which is not well constrained. Accurate measurement of emission factors (EF's) is important since EF's have long been the fundamental tool in developing national, regional, state, and local emissions inventories for air quality management decisions and in developing emissions control strategies. This work contributes to an effort to standardize the process in obtaining EF's and a great first step in developing a predictive understanding of release of particles and gases from pyrolysis. The paper is well written and brief and concise and worthy of publications with some explanations.

Biomass combustion is a complex process that consists of consecutive heterogeneous and homogeneous reactions, and a single particle of the biomass combustion process; consisting of: – heating and drying, – pyrolysis (devolatilization), – flame combustion, and – char combustion. Each process is generally believed to depend on the biomass fuel type, geometry, temperature and the combustion environment.

General comments:

The factors controlling pyrolytic properties of biomass need to be discussed in greater detail: For example, the pyrolytic properties of biomass are controlled by the chemical composition of its major components, namely cellulose, hemicelluloses, and lignin and their minor components including extractives and inorganic materials. The fuel properties and process conditions affect the pyrolysis and combustion characteristics, altering the heat generation, heat transfer and reaction rates in a complicated manner See for example [Ryu, C., et al., Effect of Fuel Properties on Biomass Combustion: Part I, Experiments – Fuel Type, Equivalence Ratio and Particle Size, Fuel, 85 (2006), 7-8, pp. 1039-1046] For a biomass with higher cellulose content, the pyrolysis rate became faster, while, the biomass with higher lignin content gave a slower pyrolysis rate, which means that pyrolysis of cellulose and hemicellulose will proceed to completion before the pyrolysis of lignin reaches a very advanced stage [see: Gani, A., Naruse, I., Effect of Cellulose and Lignin Content on Pyrolysis and Combustion Characteristics for Several Types of Biomass, Renewable Energy, 32 (2007), 4, pp. 649-661, and Roberts, A. F., A Review of Kinetics Data for the Pyrolysis of Wood and Related Substances, Combustion and Flame, 14 (1970), 2, pp. 261-272]. The cellulose and lignin content in the biomasses were one of the important parameters required to evaluate pyrolysis characteristics which suggests fuel type dependence contrary to the conclusions by the authors.

Line 139: "constant temperature experiments at 400, 500, and 600 C". The distinction between slow and fast pyrolysis has been described for example 400 and 500 ºC are considered as slow [Panwar, N. L., et al., Thermo Chemical Conversion of Biomass – Eco Friendly Energy Routes, Renewable and Sustainable Energy Reviews, 16 (2012), 4, pp. 1801-1816] pyrolysis which is an irreversible process in which thermal decomposition of the biomass material and residence time varies from 5 to 30 min in this case biomass is slowly devolatilized, and volatiles do not escape as rapidly as they do in fast pyrolysis. On the other hand, 600 ºC is considered fast pyrolysis and characterized with a short residence time, [ see: Vamvuka, D., Bio-oil, Solid and Gaseous Biofuels from Biomass Pyrolysis Processes – An Overview, International Journal of Energy Research, 35 (2011), 10, pp. 835-862, Mohan, D., et al., Pyrolysis of Wood/Biomass for Bio-oil: A Critical

Review, Energy & Fuels, 20 (2006), 3, pp. 848-889]. The authors need to clarify if their experiments correspond to fast or slow pyrolysis

Line 297-299: "We have shown that differences in yields of particles and gases could be explained by reactor temperature, moisture content, and wood size, and that time-dependent emission of products can be explained by differing thermal diffusivities rather than wood species."

It is certainly true that that wood particle pyrolysis process is dependent on its size, geometry and specific, anisotropic structure. It was shown that the chemical composition of biomass and its main constituents (lignin, cellulose, and hemicelluloses) plays an important role in the pyrolysis process. Thermal degradation of lignin begins at 200 °C and between 225 °C and 450 °C the reaction process becomes exothermic. Unless we assume that all wood samples have similar chemical composition the dependence on the wood species cannot be ignored. There are a lot of differences in the structures of hardwood and softwood, which has a large influence on the heat and mass transfer processes occurring during pyrolysis. Softwood (such as spruce) has long fibers (1.9-5.6 cm) and a more homogeneous structure, while hard wood (such as beech) has a heterogeneous structure and small fibers (0.4-1.9 cm) [Sokele, B., Wood Chemistry (in Bosnian), University of Sarajevo, Sarajevo, 197; Westerhof, R. J. M., et al., Effect of Particle Geometry and Microstructure on Fast Pyrolysis of Beech Wood, Energy Fuels, 26 (2012), 4, pp. 2274-2280]. These specifics have a strong influence on the diffusion mechanisms related to heat and species transfer trough biomass particle.

In fact, Petar M. GVERO et al Pyrolysis as a Key Process in Biomass Combustion ... THERMAL SCIENCE: Year 2016, Vol. 20, No. 4, pp. 1209-1222, showed the importance of fuel and environment characteristics on the pyrolysis and combustion process of biomass. Regarding to mass transfer in the biomass particle during pyrolysis, two types of diffusion processes occur: diffusion of the bonded moisture through cell walls, and diffusion of the gas's mixture. Diffusion is the dominant transport mechanism during pyrolysis that occurs on the lower heat fluxes and temperatures, while at higher temperatures, convection is dominant.

Additional Comments:
1. Even though citation is provided for the experimental details a little more detail would be helpful.

2. The range of temperatures selected for this experiment may not be representative temperatures in wildfires or even cookstoves. What happens at much higher temperatures corresponding to flaming fires?

3. Is there a way to relate and quantity the pyrolysis process to the commonly used MCE commonly used to determine combustion conditions?

---

## Author Comment (AC1)

**Response to Reviewer 1**

The reviewer comments are in italics and our responses are in regular font. We thank the reviewer for reading our paper and providing valuable feedback on our work.

**Comment 1**

*The factors controlling pyrolytic properties of biomass need to be discussed in greater detail: For example, the pyrolytic properties of biomass are controlled by the chemical composition of its major components, namely cellulose, hemicelluloses, and lignin and their minor components including extractives and inorganic materials. The fuel properties and process conditions affect the pyrolysis and combustion characteristics, altering the heat generation, heat transfer and reaction rates in a complicated manner. See for example [Ryu, C., et al., Effect of Fuel Properties on Biomass Combustion: Part I, Experiments Fuel Type, Equivalence Ratio and Particle Size, Fuel, 85 (2006), 7-8, pp. 1039-1046]. For a biomass with higher cellulose content, the pyrolysis rate became faster, while, the biomass with higher lignin content gave a slower pyrolysis rate, which means that pyrolysis of cellulose and hemicellulose will proceed to completion before the pyrolysis of lignin reaches a very advanced stage [see: Gani, A., Naruse, I., Effect of Cellulose and Lignin Content on Pyrolysis and Combustion Characteristics for Several Types of Biomass, Renewable Energy, 32 (2007), 4, pp. 649-661, and Roberts, A. F., A Review of Kinetics Data for the Pyrolysis of Wood and Related Substances, Combustion and Flame, 14 (1970), 2, pp. 261-272]. The cellulose and lignin content in the biomasses were one of the important parameters required to evaluate pyrolysis characteristics which suggests fuel type dependence contrary to the conclusions by the authors.*

We agree that the fuel properties affect pyrolysis rates and heat of reactions. However, in the works cited by the reviewer, this principle was only illustrated on small wood (called thermally thin) at slow heat rates in thermogravimetric analysis. In these situations, reactions control the process instead of heat transfer. The role of wood components has never been reported on large wood explicitly. The modeling approach by Park *et al.* [1] tested the importance of cellulose, hemicellulose, and lignin interactions in pyrolysis modeling of thermally thick wood and found that the model is not well described. Other approaches, including Lu *et al.* [2], Gronli *et al.* [3], Fawaz *et al.* [4] found that a lumped kinetic approach enabled the modeling of wood pyrolysis at different conditions. In Fawaz *et al.* [4], we found that a simplified kinetic pyrolysis model can predict the mass loss rate of different types of woods while only changing the physical properties of the wood. In SI section-4.2, we show the prediction of the mass loss rate of pineR and birch at 400, 500, and 600°C.

We did not include the discussion of cellulose, hemicellulose, and lignin because there is no evidence that this division alters pyrolysis in thermally thick wood. At the same time, it has been shown that heat transfer does control pyrolysis in the thermally thick wood (this is the definition of "thermally thick"). The decomposition of wood proceeds in waves; at each point in time, the pyrolysis reactions occur at an elevated temperature for all components in a finite slice of the wood. The wood components, cellulose, hemicellulose, and lignin decompose simultaneously, so there is no distinction in the decomposition behavior at elevated temperatures.

The two papers the reviewer cited were not relevant to the conditions in our work and the conditions relevant to biomass burning applications such as wildland fires and wood cookstoves. In Ryu *et al.* [5], the three wood types used were less than 35 mm long, and the temperatures reached 1200°C to study pyrolysis and combustion in a fixed bed reactor. Gani and Narsu [6] analyzed wood type at slow heating rates reaching 1173 K for 7 mg wood samples in a thermogravimeter. Similarly, the conditions are not relevant to our work, and the findings are not comparable to our work. The temperatures considered in those two papers are high (900-1400°C), and they don't represent the temperatures that favor the formation of organic aerosols. Under burning conditions, those high temperatures are associated with gas phase oxidation reactions that consume OA precursors to produce light gases. In addition, the high temperatures in the absence of oxygen promote the formation of soot. The small sizes reported in the papers do not represent the combustion of wood in real-life applications. Ryu *et al.* [5] uses several sizes in a fixed packed bed, and these conditions are not like those that occur in combustion, and Gani and Narsu [6] use wood masses that are less than 1 g. In these sizes, chemical reactions dominate pyrolysis in contrast with larger sizes (common in combustion), where pyrolysis is dominated by heat transfer.

Nevertheless, we will clarify the discussion of the wood type in section 3.4, line 299: no studies have investigated the role of the chemical makeup of the wood in the pyrolysis of thermally thick wood. The difference in real-time emissions among each group of wood (low- and high-density wood groups) can be attributed to the differences in wood composition.

**Comment 2**

*"constant temperature experiments at 400, 500, and 600C". The distinction between slow and fast pyrolysis has been described*

*for example 400 and 500C are considered as slow [Panwar, N. L., et al., Thermo Chemical Conversion of Biomass – Eco Friendly Energy Routes, Renewable and Sustainable Energy Reviews, 16 (2012), 4, pp. 1801-1816] pyrolysis which is an irreversible process in which thermal decomposition of the biomass material and residence time varies from 5 to 30 min in this case biomass is slowly devolatilized, and volatiles do not escape as rapidly as they do in fast pyrolysis. On the other hand, 600 oC is considered fast pyrolysis and characterized with a short residence time, [ see: Vamvuka, D., Bio-oil, Solid and Gaseous Biofuels from Biomass Pyrolysis Processes An Overview, International Journal of Energy Research, 35 (2011), 10, pp. 835-862, Mohan, D., et al., Pyrolysis of Wood/Biomass for Bio-oil: A Critical Review, Energy Fuels, 20 (2006), 3, pp. 848-889]. The authors need to clarify if their experiments correspond to fast or slow pyrolysis*

Slow and fast pyrolysis are defined based on a threshold heating rate in the wood. Since a thermal gradient forms between the surface and the center, the wood does not have a homogeneous heating rate. The terms slow and fast pyrolysis are applicable for small wood less than 1 mm in diameter, such as the ones used in Panwar *et al.* [7]. These terms are used to describe reactor-level bio-oil production [8] in fixed and fluidized beds. These terms cannot be easily applied to thermally thick wood. We use the term thermally thick wood to highlight the role of heat transfer and to emphasize that pyrolysis is not uniform in the wood.

In our experiments, we performed pyrolysis at 400, 500, and 600°C reactor temperatures. The heating rate at the surface ranged between 4-10 °C/sec for birch wood and 0-1 °C/sec for the wood center, among reactor temperatures. Using the thresholds defined in literature for slow and fast pyrolysis at 1 °C/sec, the pyrolysis at the surface is fast pyrolysis, and at the center is slow pyrolysis. We added birch and pineR wood heating rates values to section 3.1 line 203, and in SI section-4.2 (Figure S31-S36) we showed the real-time heating rates to emphasize the two heating regimes between the surface and center.

**Comment 3**
*Line 297-299: "We have shown that differences in yields of particles and gases could be explained by reactor temperature, moisture content, and wood size, and that time dependent emission of products can be explained by differing thermal diffusivities rather than wood species." It is certainly true that that wood particle pyrolysis process is dependent on its size, geometry and specific, anisotropic structure. It was shown that the chemical composition of biomass and its main constituents (lignin, cellulose, and hemicelluloses) plays an important role in the pyrolysis process. Thermal degradation of lignin begins at 200C and between 225C and 450C the reaction process becomes exothermic. Unless we assume that all wood samples have similar chemical composition the dependence on the wood species cannot be ignored. There are a lot of differences in the structures of hardwood and softwood, which has a large influence on the heat and mass transfer processes occurring during pyrolysis. Softwood (such as spruce) has long fibers (1.9-5.6cm) and a more homogeneous structure, while hard wood (such as beech) has a heterogeneous structure and small fibers (0.4-1.9 cm) [Sokele, B., Wood Chemistry (in Bosnian), University of Sarajevo, Sarajevo, 197; Westerhof, R. J. M., et al., Effect of Particle Geometry and Microstructure on Fast Pyrolysis of Beech Wood, Energy Fuels, 26 (2012), 4, pp. 2274-2280]. These specifics have a strong influence on the diffusion mechanisms related to heat and species transfer trough biomass particle. In fact, Petar M. GVERO et al Pyrolysis as a Key Process in Biomass Combustion ... THERMAL SCIENCE: Year 2016, Vol. 20, No. 4, pp. 1209-1222, showed the importance of fuel and environment characteristics on the pyrolysis and combustion process of biomass. Regarding to mass transfer in the biomass particle during pyrolysis, two types of diffusion processes occur: diffusion of the bonded moisture through cell walls, and diffusion of the gas's mixture. Diffusion is the dominant transport mechanism during pyrolysis that occurs on the lower heat fluxes and temperatures, while at higher temperatures, convection is dominant.*

We agree with the reviewer that the fuel and environmental characteristics influence pyrolysis. The reviewer mentions microstructural variables that occur within a wood sample. These microstructural variables may be influential for small wood (thermally thin wood). However, the studies that report these influences did not investigate macro-scale variables such as wood size, pyrolysis temperature, wood type, and moisture content that dominate heat and mass transport within thermally thick wood.

In this work and a previous publication, we have demonstrated that macro-scale phenomena are sufficient to explain the major features of pyrolysis behavior that affect emissions to the atmosphere. In the present paper, we showed that wood type does affect these macro-scale variables, but mainly because of the wood density. We used real-time measurements of mass loss rates and emissions (this paper), combined with modeling approaches [4]. Although we agree that wood composition and other variables may affect the release, the purpose of this work is to identify the significant factors that influence emission, which we have done without relying on the type of compositional breakdown described by the reviewer.

We will change this sentence in the conclusion of section 3.4 in line 309 that the reviewer highlighted to reflect the conciseness of our work, as follows: We have shown that differences in yields of particles and gases could be explained by reactor temperature, moisture content, and wood size, and that the difference in real-time emission among wood types can be explained by the difference in the physical properties of the wood.

In the following text, we will briefly discuss some of the issues raised in this comment.

*Microstructure and wood type:* The permeability of char is the critical variable that affects pyrolysis and material flow out of the solid matrix. The microstructure of hardwoods and softwoods does not influence the flow of materials because the flow occurs from the production site through the char to the surrounding environment. Regardless of the initial microstructure, the permeability of char is significantly higher than that of wood. In addition, we show in section 3.2 that wood size is essential in determining the total amount of material released from the wood at low pyrolysis temperature.

*Mass transfer:* The mass transfer is predominantly convective flow in a porous matrix. The flow is caused by the pressure evolution in the wood from the formation of gaseous products from pyrolysis. The flow aspect was measured [9, 10] and modeled [11, 12] in literature. In pyrolysis modeling, mass transfer of volatile pyrolysis products and water vapor is based on gas flows driven by pressure gradients, and the mass flux is solved using Darcy's law. Within Gvero *et al.* [13] on page 7, the importance of pressure evolution is discussed for different wood orientations, sizes, and shapes. Thus, we disagree with the reviewer that diffusion is dominant in these situations, and our position is supported even by the citation given by the reviewer.

**Short Comments**
*Even though citation is provided for the experimental details a little more detail would be helpful.*
We changed the paragraph in line 90 and added 138 words to provide more detail on the experiments.

*The range of temperatures selected for this experiment may not be representative temperatures in wildfires or even cookstoves. What happens at much higher temperatures corresponding to flaming fires?*
The experiments were performed at temperatures favorable for particle formation. At high temperature pyrolysis conditions, secondary pyrolysis (T>600°C) may consume OA precursors to form light gases such as CO, $CO_2$, and small hydrocarbons and tertiary pyrolysis (T>900°C) may consume OA precursors to form soot. In the presence of a flame, OA precursors are consumed in the oxidation reactions to become light gases such as CO, $CO_2$, and water vapor. We have added this explanation to section 2.3 line 142, as follows: This range was selected because at temperatures less than 400°C pyrolysis was extremely slow and did not occur at certain conditions, and at temperatures higher than 600°C secondary reactions consumed OA precursors before formation. In combustion conditions, temperatures higher than 600°C are associated with flaming combustion.

*Is there a way to relate and quantity the pyrolysis process to the commonly used MCE commonly used to determine combustion conditions?*
We measured both CO and CO2 concentrations in all experiments and we put a representative plot in SI section 4.2 Figure S24 to show the change in MCE at the three temperatures for birch.

**References**

[1] Won Chan Park, Arvind Atreya, and Howard R. Baum. Experimental and theoretical investigation of heat and mass transfer processes during wood pyrolysis. *Combustion and Flame*, 157(3):481–494, 2010.

[2] Hong Lu, Warren Robert, Gregory Peirce, Bryan Ripa, and Larry L. Baxter. Comprehensive Study of Biomass Particle Combustion. *Energy & Fuels*, 22(4):2826–2839, 2008.

[3] Morten G. Grønli and Morten C. Melaaen. Mathematical Model for Wood Pyrolysis-Comparison of Experimental Measurements with Model Predictions. *Energy & Fuels*, 14(4):791–800, 2000.

[4] Mariam Fawaz, Chris Lautenberger, and Tami C Bond. Prediction of organic aerosol precursor emission from the pyrolysis of thermally thick wood. *Fuel*, 269:117333, 2020.

[5] Changkook Ryu, Yao Bin Yang, Adela Khor, Nicola E Yates, Vida N Sharifi, and Jim Swithenbank. Effect of fuel properties on biomass combustion: Part i. experiments—fuel type, equivalence ratio and particle size. *Fuel*, 85(7-8):1039–1046, 2006.

[6] Asri Gani and Ichiro Naruse. Effect of cellulose and lignin content on pyrolysis and combustion characteristics for several types of biomass. *Renewable energy*, 32(4):649–661, 2007.

[7] NL Panwar, Richa Kothari, and VV Tyagi. Thermo chemical conversion of biomass–eco friendly energy routes. *Renewable and Sustainable Energy Reviews*, 16(4):1801–1816, 2012.

[8] Andres Anca-Couce. Reaction mechanisms and multi-scale modelling of lignocellulosic biomass pyrolysis. *Progress in Energy and Combustion Science*, 53:41–79, 3 2016.

[9] Calvin K. Lee, Robert F. Chaiken, and Joseph M. Singer. Charring pyrolysis of wood in fires by laser simulation. *Symposium (International) on Combustion*, 16(1):1459–1470, 1977.

[10] Bertil Frendlund. *A model for heat and mass transfer in timber structures during fire: a theoretical, numerical and experimental study*. PhD thesis, Lund University, 1988.

[11] Edward J. Kansa, Henry E. Perlee, and Robert F. Chaiken. Mathematical model of wood pyrolysis including internal forced convection. *Combustion and Flame*, 29(C):311–324, 1977.

[12] Kenneth Bryden, Kenneth Ragland, and Christopher Rutland. Modeling thermally thick pyrolysis of wood. *Biomass and Bioenergy*, 22(1):41–53, 2002.

[13] Petar M Gvero, Sasa Papuga, Indir Mujanic, and Sran Vaskovic. Pyrolysis as a key process in biomass combustion and thermochemical conversion. *Thermal Science*, 20(4):1209–1222, 2016.

---

## Author Comment (AC2)

**Response to Reviewer 2**
* * *
The reviewer comments are in italics and our responses are in regular font. We thank the reviewer for reading our paper and providing valuable feedback on our work.

**Comment 1**

*Page 4 Line 96: It is not specified (not even in Table S1) at what sampling temperature aerosol particles are measured inside the second dilution line. Temperature strongly affects particle size distribution of OA because the abundance of semi-volatile compounds. If it is near ambient, the experiments represent a sort of theoretical upper limit for the emission of OA particles into the atmosphere, because all other processes that are admittedly not considered would reduce OA emissions and increase the emission of permanent gases and soot.*

The sampling was done at near ambient temperature during the experiments. We added that to section 2.1 line 106. The possible transformations of the OA precursors are discussed in the introduction line 35- 44. In this work, the conditions selected allow for two transformations of the precursors, OA formation and secondary pyrolysis reactions. At 600°C, secondary pyrolysis reactions may consume OA precursors at 400 and 500°C the precursors may form OA without any other transformations. With the sampling at ambient conditions, OA can be formed, and the yield of OA at the end of the experiment can be the maximum yield that wood can form OA. We addressed that in the results section 3.1 line 230.

**Comment 2**

*Page 4 Line 103: The reported observation that "a dark, heavy, sticky, material was released in some experiments" is very valuable because it indicates that formation of tar balls, a very significant subgroup of brown carbon particles in the atmosphere. While I understand that this compounds cannot be collected as particles due to experimental limitations, it might be very useful if their presence were indicated e.g. in Table 1 either with Yes/No or with some semi-quantitative visual classification (e.g. +, ++, +++) if they can be reconstructed from the experimental records.*

This request has been added to table 1.

**Comment 3**

*Page 4 Lines 258–260: These seemingly contradicting findings may be rationalized by the potentially different chemical compositions of organic compounds released when water is present due to supplementary processes such as steam distillation. However, it seems that on a mass basis energy reduction by the presence of water predominates in all cases observed.*

The chemical composition of the particles is beyond the scope of the paper. Without more information on the conditions of the experiments in each paper we can't attribute to the difference in emission to energy reduction or chemical composition of the particles. However, our work shows that during pyrolysis particle emission is reduced in woods with high moisture content compared to woods with low moisture content.

**Comment 4**

*Page 4 Lines 112–115: Names of tree species must be written in italics*

This has been changed.

**Comment 5**

*Page 6 Line 161: Typographical errors: the character En dash should be used here and the symbol  instead of \**

This has been changed.